# Distinct Effects of Moxifloxacin and Bedaquiline on Growing and ‘Non-Culturable’ *Mycobacterium abscessus*

**DOI:** 10.3390/microorganisms11112690

**Published:** 2023-11-02

**Authors:** Andrey L. Mulyukin, Deborah Recchia, Nadezhda A. Kostrikina, Maria V. Artyukhina, Billy A. Martini, Alessandro Stamilla, Giulia Degiacomi, Elena G. Salina

**Affiliations:** 1Winogradsky Institute of Microbiology, Research Center of Biotechnology of the Russian Academy of Sciences, 119071 Moscow, Russianadin-kost@yandex.ru (N.A.K.); 2Department of Biology and Biotechnology “Lazzaro Spallanzani”, University of Pavia, 27100 Pavia, Italyalessandro.stamilla@unipv.it (A.S.);; 3Bach Institute of Biochemistry, Research Center of Biotechnology of the Russian Academy of Sciences, 119071 Moscow, Russia; masha.artyuhina@yandex.ru (M.V.A.);

**Keywords:** *Mycobacterium abscessus*, non-culturability, viability, drug tolerance, bactericidal activity

## Abstract

*Mycobacterium abscessus* has recently emerged as the cause of an increasing number of human infections worldwide. Unfortunately, it is highly resistant to existing drugs, and new specific agents to combat *M. abscessus* have not yet been found. The discovery of antibiotics that are effective not only against replicating but also against dormant and often recalcitrant cells is a daunting challenge. In this study, we developed a model of non-replicating *M. abscessus,* which represents a valuable screening tool for antibacterial agents. Thus, we demonstrated that, under a deficiency of potassium ions in the growth media and prolonged incubation, *M. abscessus* entered a ‘non-culturable’ state with a significant loss of colony-forming ability, but it retained viability, as confirmed using the most-probable-number (MPN) assay. The ‘non-culturable’ mycobacteria possessed decelerated cellular metabolism and noticeable differences in cell morphology from actively growing mycobacteria. ‘Non-culturable’ cells were used in a comprehensive screening of the efficacy of antibiotics, along with actively growing cells. Both CFU and MPN tests confirmed the prominent bactericidal effect of moxifloxacin on actively growing and ‘non-culturable’ *M. abscessus,* as proven by less than 0.01% of cells surviving after antibiotic treatment and prolonged storage. Bedaquiline exhibited a comparable bactericidal effect only on metabolically inactive non-culturable cells aged for 44 days. There were reductions ranging from 1000 to 10,000-fold in CFU and MPN, but it was not so efficient with respect to active cells, resulting in a bacteriostatic effect. The demonstrated specificity of bedaquiline in relation to inert non-replicating *M. abscessus* offers a new and unexpected result. Based on the findings of this research, moxifloxacin and bedaquiline can be regarded as potential treatments for infections caused by *M. abscessus*. In addition, a key outcome is the proposal to include the combination of viability assays for comprehensive testing of drug candidates. Relying on CFU-based assays alone resulted in overestimates of antibacterial efficacy, as demonstrated in our experiments.

## 1. Introduction

Non-tuberculous mycobacteria (NTM) have become a growing global concern due to increasing numbers of infected individuals [1,2], particularly those with cystic fibrosis, bronchiectasis, chronic obstructive pulmonary disease, and previous tuberculosis infection. NTM comprise around 200 species to date [3], 95% have been isolated from different environments, being saprophytes or non-pathogenic. Only a few species cause serious and often opportunistic infections in humans, including the *Mycobacterium abscessus* complex (MABSC). MABSC comprises three subspecies: *M. abscessus* subsp. *abscessus* (*M. abscessus*), *M. abscessus* subsp. *bolletii*, and *M. abscessus* subsp. *massiliense*. Among NTM, *M. abscessus* is the most commonly identified species responsible for severe respiratory, skin, and mucosal infections in humans. MABSC accounts for more than 90% of cystic fibrosis cases along with the slowly growing *Mycobacterium avium* complex [2,4,5]. *M. abscessus* has a high level of resistance to antibiotics [1,6], including macrolides, aminoglycosides, rifamycins, tetracyclines, and β-lactams. The resistance mechanisms involve mutations in the genes encoding molecular targets of drugs, intrinsic resistance or tolerance, the low permeability of the cell wall [7], and activation of drug efflux pumps, as well as deficient drug-activating enzymes that no longer convert prodrugs into active metabolites and the presence of numerous enzymes that neutralize drugs or modify their specific targets [6,8].

Combatting *M. abscessus* infections relies on long-term antimicrobial therapy with significant side effects and high failure rates of 50% or more [9,10,11]. Although the Cystic Fibrosis Foundation of the United States and the European Cystic Fibrosis Society have published consensus recommendations for the treatment of NTM in patients with cystic fibrosis, there is currently no single standard treatment regimen for MABSC infections [12]. For the treatment of NTM, the official clinical practice guidelines from ATS/ERS/ESCMID/IDSA [13] recommend a combination therapy consisting of three or more active drugs, such as tigecycline, imipenem, cefoxitin, and amikacin, for up to 24 months. In addition, many guidelines recommend continued treatment for 12 months after a negative sputum culture [10,14]. New powerful drugs to treat *M. abscessus* infections have not yet been found despite manifold efforts [10,15]. Drug candidates are commonly tested on active bacteria grown in nutrient-rich or balanced synthetic media; that is, in situations that rarely occur in nature, leading to biased outcomes in the process of drug discovery. In terms of clinical relevance, it should be emphasized that mycobacteria, like other microorganisms, can persist under conditions that differ from those present in aerated, nutrient-rich broth typically used to grow test cultures for MIC assays [16,17]. Changes in nutritional conditions or starvation can cause the entry of bacteria to a dormant or non-replicating state. Therefore, in the search for new antibacterial agents, it is critical to include testing of drug candidates on metabolically inert, often recalcitrant cells [18,19], which do not grow and multiply or acquire a transiently ‘non-culturable’ state.

Previous studies have demonstrated differences in the susceptibility of *Mycobacterium tuberculosis* to new drug candidates, depending upon the physiological state of the cells (active versus inactive) [19,20]. Benzothiazinone-043 (BTZ043TB) and piperazinobenzothiazinone (PBTZ169)—the anti-tuberculosis agents that effectively inhibit the biosynthesis of arabinogalactan, a key component of the mycobacterial cell wall—displayed efficacy on actively dividing *M. tuberculosis* cells only [21,22]. Pretomanid (PA-824) was bactericidal both for replicating and non-replicating *M. tuberculosis* in low-oxygen conditions; however, minimal inhibitory concentrations (MIC) and minimal bactericidal concentrations (MBC) were much higher for non-replicating than active mycobacteria [23,24]. Pretomanid inhibited the cell wall biosynthesis via blockage of the oxidation of hydroxymycolate to ketomycolate; under anaerobic conditions, pretomanid caused respiratory poisoning of the bacterial cells through the release of reactive nitrogen species [25]. In contrast, bedaquiline (TMC207), an ATP synthase inhibitor, effectively killed both replicating and dormant *M. tuberculosis* cells with the same target specificity [26].

Dormant cells of *M. abscessus* have found limited application in tests for the bactericidal activity of potent drugs or their combinations. Particularly, *M. abscessus* cells, which underwent starvation in PBS [16], or survived under oxygen deprivation and nutrient starvation in a non-replicative state [17], or subjected to hypoxic conditions [27], have been the main targets for assessing bactericidal activity. Entry into the dormant state with the loss of the ability to form colonies (transiently ‘non-culturable’ state) is commonly and reproducibly attainable due to a deficiency of potassium ions in the culture medium, as previously demonstrated for other mycobacteria, notably, *M. tuberculosis* [28,29,30]. However, this has not been shown for *M. abscessus*, which is likewise clinically significant.

The main objective of this study was to assess the efficacy of antibiotics against active and dormant *Mycobacterium abscessus* cells. To this aim, we investigated the capability of this bacterium to enter a metabolically inactive state under the deficiency of potassium ions in the growth media and focused on developing a model of non-replicating cells as a useful tool for the screening of antibacterial agents. This study included testing a combined approach to monitor the residual viability of both active and dormant *M. abscessus* cells after exposure to antibacterial agents screened for a reliable evaluation of their bacteriostatic and bactericidal effects. Finally, we attempted to compare the efficacy of several antibiotics, particularly anti-tuberculosis agents, with respect to actively growing and non-replicating *M. abscessus*.

## 2. Materials and Methods

Bacterium and media. *M. abscessus* ATCC 19977^T^ was provided by European Polytechnic School of Lausanne (Lausanne, Switzerland) and stored at −70 °C. Starter cultures were initially grown in Sauton medium, containing: KH_2_PO_4_, 0.5 g; MgSO_4_·7H_2_O, 1.4 g; L-asparagine, 4 g; glycerol, 60 mL; ferric ammonium citrate, 0.05 g; sodium citrate, 2 g; 1% ZnSO_4_ · 7H_2_O, 0.1 mL; H_2_O, to 1 L; pH 7.0 (adjusted with 1 M NaOH) with addition of ADC (albumin-dextrose-catalase) growth supplement [31] (Himedia, Mumbai, India) and 0.05% Tween-80 (Neofroxx GmbH, Einhausen, Germany) at 37 °C with shaking at 200 rpm for 10 days.

To obtain transiently ‘non-culturable’ cells, a starter culture was inoculated (0.25%) into potassium-free Sauton media with ADC and Tween-80 in which K^+^ ions were equimolarly substituted for Na^+^ ions [29]. Cultures were incubated in the K^+^-deficient medium in loose-capped flasks at 37 °C with shaking at 200 rpm for 45 days.

Viability tests. Tenfold serial dilutions of *M. abscessus* cultures were plated in triplicates onto agar-solidified Sauton medium with ADC in Petri dishes and incubated at 37 °C for 5 days followed by CFU counting. The same dilutions were inoculated in 48-well Corning plates with the liquid Sauton medium to evaluate the most probable number of viable cells (MPN) after incubation without shaking at 37 °C for 10 days. Wells with visible bacterial growth were counted as positive, and MPN values were calculated using standard statistical methods [32].

Respiratory assay. Respiratory activity of cells was assessed by measuring their ability to reduce 2,6-dichlorophenol-indophenol sodium salt (DCPIP) (Merck, Darmstadt, Germany) in the presence of menadione [33]. The reaction mixture (200 µL) contained 0.5 mM DCPIP; 0.15 mM menadione, and *M. abscessus* cell suspension (1 × 10^7^ of viable cells, estimated by MPN assay) in 0.01 M phosphate buffer pH 7.0. De-colorization of DCPIP upon reduction was monitored at λ = 600 nm using a Fluostar Omega spectrofluorometer (BMG-Labtech, Ortenberg, Germany). The respiratory activity was expressed in mol of reduced DCPIP per cell per minute as the average of 3 measurements with the relative errors < 5%.

Thermal stability. Cells from actively growing and aged cultures (24 and 44 days) in K+-deficient medium were transferred to sterile tubes, washed, and resuspended in their own supernatants to OD_600_ = 0.1. The suspensions were heated for 20 min at different temperatures (50–80 °C) without agitation. The numbers of viable cells in the heated suspensions were determined using CFU and MPN counting as described above.

Transmission electron microscopy. Cells from cultures were pelleted (centrifugation 4000× *g*) and fixed in 2.5% (*w/v*) in 0.1 M sodium cacodylate buffer (pH 7.2) for 2.5 h and then in 1% (*w*/*v*) osmium tetroxide. After dehydration in ethanol solutions, the fixed material was embedded in epoxy resin (kit CAS 45359-1EA-F, Sigma, New York, NY, USA), sectioned on an ultratome (LKB, Stockholm, Sweden), and stained with lead citrate. Thin sections were examined under a JEM-1400 electron microscope (Jeol, Tokyo, Japan). Electron microscopy was performed in UNIQEM Collection Core Facility.

Bactericidal effect. Both ‘non-culturable’ cells (K^+^-deficient medium supplemented with ADC and Tween-80, aged for 24 and 44 days) and actively growing cells, as the control, (Sauton medium supplemented with ADC and Tween-80, 3 days) were washed with fresh medium, diluted with their own supernatant to OD_600_ = 0.1, and challenged to 100 µg/mL of moxifloxacin (Moxi) (Merck, Germany), or 200 µg/mL of bedaquiline (BDQ) (Biosynth, USA), or rifampicin (RIF) (Merck, Germany) for 15 min (day 0), 7, 14, and 21 days (37 °C; agitation, 200 rpm). The used concentrations of antibiotics corresponded to 50 minimal inhibitory concentrations (MIC) determined in primary screening tests based on incubation of active *M. abscessus* cultures using the resazurin microtiter assay (REMA), as previously described [34]. Our MIC tests exploited also amikacin (Amk), clofazimin (CFZ), and linezolid (LNZ), but they were less active than Moxi, BDQ, and RIF and were not taken for further studies. Cultures were washed with fresh Sauton medium supplemented with ADC and Tween-80 to remove antibiotics. Then, the bactericidal effect of the compound on the mycobacteria under study was assessed by CFU and MPN tests.

Statistical analysis was performed using Microsoft^®^ Office^®^ Excel 2016 MSO (16.0.4639.1000). The data were expressed as the mean ± standard deviation. Three independent experiments were performed. Data were analyzed using Student’s *t*-test; *p* < 0.05 was considered statistically significant.

## 3. Results

### 3.1. K^+^ Sequestration Causes M. abscessus to Enter a Metabolically Dormant State with a Loss of Colony-Forming Ability

During prolonged incubation of post-stationary-phase *M. abscessus* cultures that were grown in the K^+^-deficient Sauton medium, cells showed a reduced ability to form colonies on nutrient agar, while the MPN values were stable, indicating that cells remained viable and could grow in the standard liquid medium. Notably, the proportion of cells, classified as transiently ‘non-culturable’, increased from 90% at day 24 to 98% at day 44 with aging of culture (as judged from CFU: MPN ratios) (Figure 1A). Correspondingly, the reduced respiratory activity of cells in 24-day and 44-day-old cultures, as judged from tests with DCPIP (Figure 1B), pointed to the decelerated cellular metabolism, consistently with at least most cells being in a metabolically inert state. Both actively growing cells and cells in a 24-day-old culture subjected to K^+^ deficiency displayed similar thermal resistance profiles (Figure 2A,B). Cells in 44-day-old K^+^-deficient cultures were not superior to active cells in thermal stability judging from both CFU and MPN values.

TEM analysis showed noticeable differences in both the integrity and morphology of *M. abscessus* cells in actively growing culture and old culture (aged for 44 days) under K^+^ deficiency. Most mycobacterial cells in the control culture were intact, while some underwent division by septation (Figure 3A). All cells possessed compact and dense cell walls, intact cytoplasmic membranes, and finely grained cytoplasm with numerous membrane-like structures and dark granules (Figure 3B).

The mycobacterial population in a 44-day-old culture was heterogeneous in the cell morphology. Intact cells possessed a granular texture of the cytoplasm with numerous electron-dense and opaque inclusions and a discernible nucleoid (Figure 3C). The cell wall of intact cells in K^+^- sequestered cultures was thicker and more loosened than in the case of active cells (Figure 3E,F). Consistently with a slightly reduced MPN number at day 44 (Figure 1A), moribund or dead cells with damaged cell walls and destructed intracytoplasmic components were present in old cultures (Figure 3D). Hence, intact *M. abscessus* cells in transiently ‘non-culturable’ populations differed in morphology from actively growing mycobacteria.

Finally, we were able to obtain *M. abscessus* cultures with metabolically inert cells that lost their colony-forming ability. These cultures were used as a proof-of-concept to screen antimicrobial agents.

### 3.2. Rifampicin, Moxifloxacin, and Bedaquiline Exert Different Effects on Growing and Metabolically Inert M. abscessus Cells: Results from Commonly Applied CFU Tests

In the absence of novel compounds for treating *M. abscessus* infections, repurposing the existing antibacterial agents, primarily used against *M. tuberculosis*, seems to be a good strategy. We found that, after the addition of rifampicin (RIF) or bedaquiline (BDQ) to cultures with actively growing cells, the CFU numbers were almost the same as at day 0, within 15 min of challenge to antibiotics, and the number of cells with the preserved colony-forming ability remained constant over the 7-to-21-day period following the antibiotic exposure. On the contrary, the effect of moxifloxacin (Moxi) on dividing *M. abscessus* was accompanied by a sharp decrease in CFU numbers up to three logs after prolonged (21 days) incubation of a Moxi-treated culture (Figure 4A).

RIF was not efficient against metabolically inactive cells in old post-stationary-phase cultures grown in a K^+^-deficient medium: the CFU numbers were almost the same as in the case of the control untreated cultures. Both Moxi and BDQ caused a decrease in CFU titers by two or more orders, and this effect was strengthened with further incubation of antibiotic-challenged cultures. Noteworthy, cells from 44-day-old cultures in K^+^-deficient medium more rapidly lost the ability to form colonies with time (14–21 days) after challenge to Moxi or BDQ (Figure 4C) than younger ‘non-culturable’ cells (Figure 4B).

The plateau in CFU numbers (Figure 4A), could be indicative of bacteriostatic effects of RIF and BDQ on actively growing cells. RIF exerted the same preventive effect on cells in K^+^-deficient culture, while BDQ caused the reduction in CFU numbers by more than 99% for both 24- and 44-day-old cultures in K^+^-deficient, thus displaying a strong antibacterial effect of BDQ on ‘non-culturable’ cells (Figure 4B,C). Drops in the CFU numbers by 99% to 99.99%, as estimated from declines by 2–4 decimal orders (Figure 4), could point to the bactericidal effects of Moxi on both replicating and ‘non-culturable’ cells. However, the observed discrepancy between CFU and MPN numbers for metabolically inert *M. abscessus* (Figure 1A) and the aggravation of non-culturability after challenge to antibiotics (Figure 4) prompted us to carry out additional experiments.

### 3.3. Putative or Confirmed Bactericidal Activity against M. abscessus: Findings from MPN Assays and Comparisons with CFU

To prove the bactericidal activity, we conducted MPN assays on the control and treated cultures, after discarding the medium with antibiotics and re-suspending in fresh growth-supporting Sauton medium. MPN numbers for RIF-treated actively growing cultures were similar to those at day 0, within 15 min of challenge to antibiotics (Figure 5A). MPN numbers for RIF-treated ‘non-culturable’ cells in 24- and 44-day-old cultures remained unchanged after storage within 21 days (Figure 5B,C). Thus, RIF was inefficient on both active and ‘non-culturable’ cells stored for 24 and 44 days: MPN counts were the same as at day 0 of antibiotic challenge (Figure 5B,C). So, RIF exerted a mild bacteriostatic effect, with no evidence of bactericidal activity.

With regard to Moxi, the MPN assays showed the prominent effect of this antibiotic on actively growing mycobacteria, particularly evident over a 14-to-21-day period post-antibiotic exposure according to CFU counts. The effect was not observed during the earlier stages following Moxi addition (Figure 5A). Moxi exerted a potent impact on ‘non-culturable’ metabolically inert cells. After the removal of the antibiotic and the addition of fresh medium, only a minor fraction of cells could resume growth (Figure 5B,C). Both MPN and CFU tests confirmed the enhancement of the antibacterial effect with the aging of antibiotic-exposed cultures. Thus, less than 0.01% of *M. abscessus* cells in K^+^-deficient culture (aged for 44 days) remained viable after the treatment with Moxi and further incubation, as determined using MPN assay (Figure 5C).

Conversely, when added to actively growing cultures, BDQ caused no significant changes in the number of viable cells: the MPN numbers were comparable to those of the control (Figure 5A). Both MPN and CFU assays validated the potent and relatively rapid action of BDQ against non-growing *M. abscessus* developed and incubated in a K^+^-deficient medium for an extended 44-day period. Therefore, BDQ shows promise as a candidate for targeted killing of aged ‘non-culturable’ *M. abscessus* cells.

## 4. Discussion

Expectedly, in response to K^+^-sequestration in the growth medium, *M. abscessus* adopted the metabolically inert state with the significant loss of colony-forming ability as already demonstrated for other mycobacteria [28,29]. The major difference for the mycobacteria under study was the less profound decline in CFU numbers and the higher CFU/MPN ratio than in the case of *M. smegmatis* and *M. tuberculosis* [20,28,30]. On the whole, actually viable (despite their lost colony-forming ability), non-replicating *M. abscessus* cells that were cultivated under K^+^ deficiency with decelerated respiratory activity and specific morphological traits can be regarded as an example of transitional dormancy, unlike resistant and differentiated resting cells produced under different and particular conditions [35].

This notwithstanding, cells with absent CFU capability but quite detectable in MPN tests, which were characterized in this study, can be a useful object for the screening of antibacterial agents. On the one hand, ‘non-culturable’ or dormant mycobacteria are of clinical relevance, besides actively growing cells, and may occur in clinical specimens [19,36]. On the other hand, dormant mycobacterial cells may escape from routinely used CFU-based tests due to lost ability to form colonies. Next, special and often sophisticated procedures to recover cells with transient ‘non-culturability’ to growth are difficult to implement in work with numerous agents and samples. Hence, the developed model of non-replicating *M. abscessus* can be included for further searches for known and new agents.

MIC assays are not enough to evaluate the bactericidal activity of tested agents against *M. abscessus*, and viability counting after challenge to antibacterial agents is essential to avoid an overestimation of their effects, as already recommended [16,17,27] and shown in this study. Particularly, both CFU and MPN tests are essential to monitor antibacterial effects of a tested agent in light of further development of non-culturability, as observed in our experiments for some antibiotics at least. It should be stressed here that the entry of different bacteria to the viable-but-non-culturable (VBNC) state after treatments with antibiotics is well known, and the similarities and differences between VBNC and antibiotic persister (AP) strategies are discussed in [37,38,39,40].

Based on the reported results, we believe that for estimation of the efficacy of drugs, it is necessary to monitor the ability of cell growth using an MPN assay since a subpopulation that cannot be cultured on agar media but regains growth in liquid media may exist or even enlarge in number over prolonged incubation periods after challenge to a tested antimicrobial agent. Only CFU assays may not be enable to distinguish between a killing and a VBNC-aggravating effect. If cells with lost colony-forming ability have remained viable (to be proved in additional assays) after exposure to a tested agent, their efficacy appears to be false-positive. Indeed, the combination of CFU, MPN, and the time to positivity (TTP) approach enabled a subpopulation of *M. tuberculosis* to be identified that cannot be cultured on solid media and provided convincing results on the effect of tested antibacterial agents more thoroughly [19,41]. To our knowledge, the successful use of a similar combined approach for *M. abscessus* has hitherto been described by Lanni and co-workers only [27].

Since specific agents to fight against *M. abscessus* infections are still absent, repurposing the use of already-known anti-*M. tuberculosis* antibiotics was an unavoidable choice. Our results proved that the sensitivity to anti-TB agents depended on the physiological state of *M. abscessus* cultures. Thus, nearly all tested antibiotics, effective on actively growing cultures, were inefficient toward non-replicating starved cells, except for amikacin, kanamycin, and niclosamide; and the general inactivity of tested compounds against non-replicating mycobacteria might impact in the failure of these drugs to adequately treat patients [16]. The other study showed that a few of the drugs displayed bactericidal activity even on growing *M. abscessus* cultures in the standard assay with aeration and a rich medium, but even this small set of antibiotics turned out to be inefficient on non-replicating *M. abscessus* under oxygen and nutrient limitations, or in vitro biofilm [17,42].

As a rule, non-replicating viable *M. abscessus* cells showed tolerance to individual antibiotics [16,17], as shown for *M. tuberculosis* [43]. The killing effect with respect to both growing and non-replicative *M. abscessus* cells under hypoxic cultures, as judged from a lack of regrowth in solid and liquid media, was achieved using the drug combinations BDQ-amikacin-rifabutin-clarithromycin-nimorazole or BDQ-amikacin-rifabutin-clarithromycin-metronidazole-colistin, and this bactericidal effect [27].

In this work, we demonstrated a relatively slight bacteriostatic effect of RIF on both active and ‘non-culturable’ *M. abscessus*, and this is quite explainable based on the identified resistance mechanisms [1,44]. We have found that Moxi can be an attractive antibacterial agent as it has a strong bactericidal effect against both active and ‘non-culturable’ cells in the current study. Previously, Moxi was reported to exert good activity in vitro [45] and was recommended for a possible antibiotic regimen to treat adults with *M. abscessus* disease [46]. However, its activity in *M. abscessus* zebrafish in vivo model was found to be quite limited [47]. Unexpectedly, BDQ was found to be a true killer of long-cultured metabolically inert *M. abscessus* with lost colony-forming ability grown under K^+^ deficiency, not of actively growing cells. Earlier, BDQ was proposed as a potential agent in the treatment of *M. abscessus* infections [48,49]; however, this proposal was based on prominent MIC values only. Later, BDQ was reported to be effective in the combination therapy for treatment of disseminated NTM infection in patients co-infected with HIV [50]. Yet, it is not simple to explain the demonstrated selective BDQ effect on old ‘non-culturable’ *M. abscessus* cells under study, although it can be associated with alterations in the structure of the cell wall with its loosening. This is a possible but not proven scenario, and further studies are necessary to gain more insight into mechanisms underlying the specific action toward non-replicative dormant cells.

## 5. Conclusions

The used combined approach that relied on both MPN and CFU counting, the inclusion of non-growing metabolically inert *M. abscessus* cells, and monitoring of residual viability over prolonged time can be promising for comprehensive testing of drug candidates to evaluate their potential for clinical applications. RIF, a common anti-tuberculosis agent, is not too efficient against *M. abscessus*, exerting a bacteriostatic action and preventing cell growth and multiplication. Moxi demonstrated a robust bactericidal effect against both actively growing and non-culturable *M. abscessus* obtained under a K^+^ deficiency. This proves the effectiveness of Moxi in treating infections caused by *M. abscessus*. BDQ exclusively killed aged non-culturable K^+^-depleted *M. abscessus.* The combined use of viability tests is crucial to avoid an overestimation of the efficacy of antibacterials.

## Figures and Tables

**Figure 1 microorganisms-11-02690-f001:**
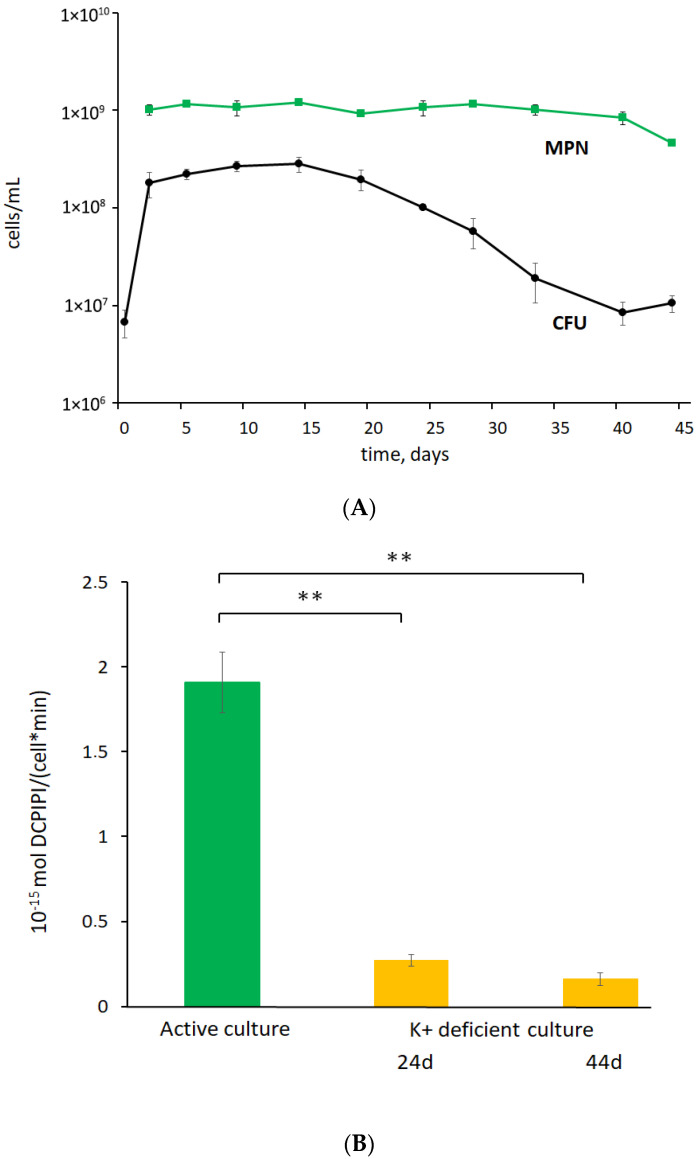
Development of non-culturability of *M. abscessus* under K^+^ deficiency: (**A**) *M. abscessus* lost their colony-forming ability after cultivation in the K^+^-deficient Sauton medium (CFU values) and could grow in the fresh liquid medium (MPN values). The experiment was repeated three times with similar results, a typical experiment is presented; (**B**) reduced respiratory activity on cells in K^+^-deficient culture. ** *p* < 0.01, unpaired *t*-test.

**Figure 2 microorganisms-11-02690-f002:**
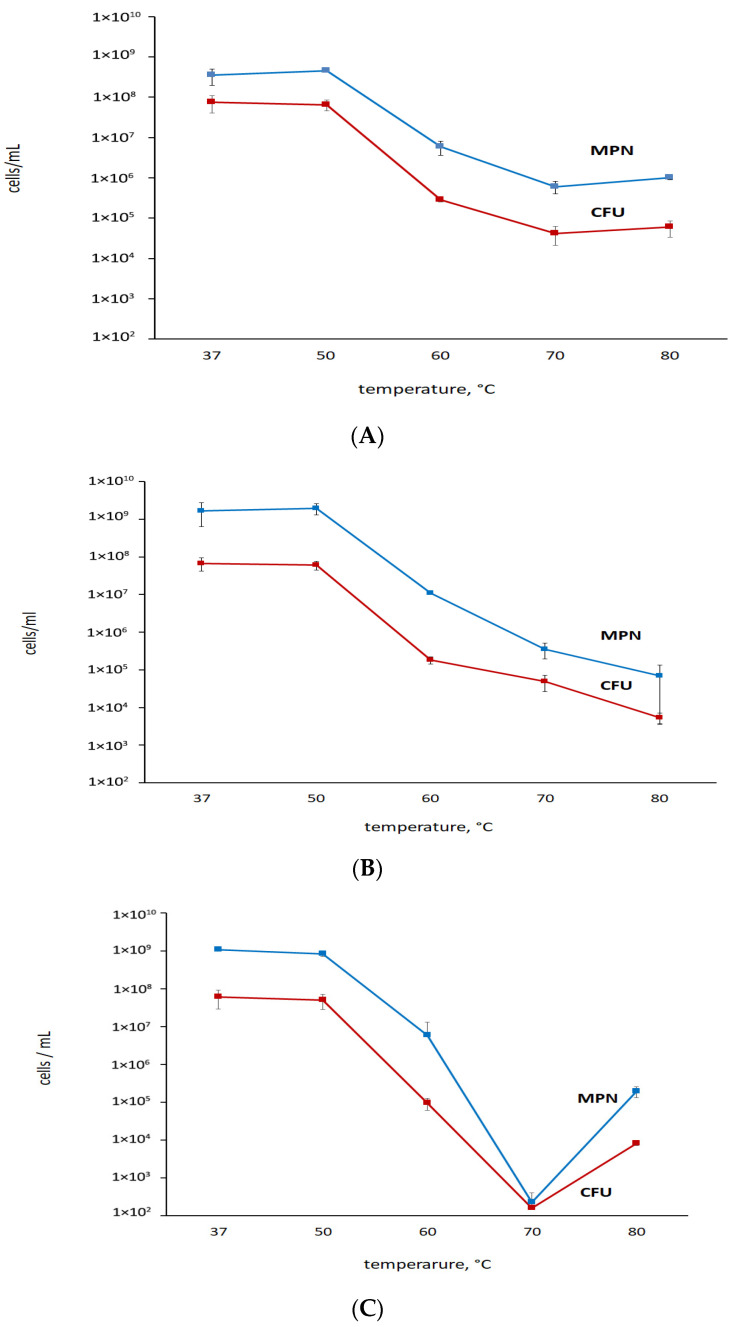
Thermal resistance of (**A**) actively growing cells and (**B**) 24-day and (**C**) 44-day-old ‘non-culturable’ *M. abscessus* cells incubated under K^+^ deficiency.

**Figure 3 microorganisms-11-02690-f003:**
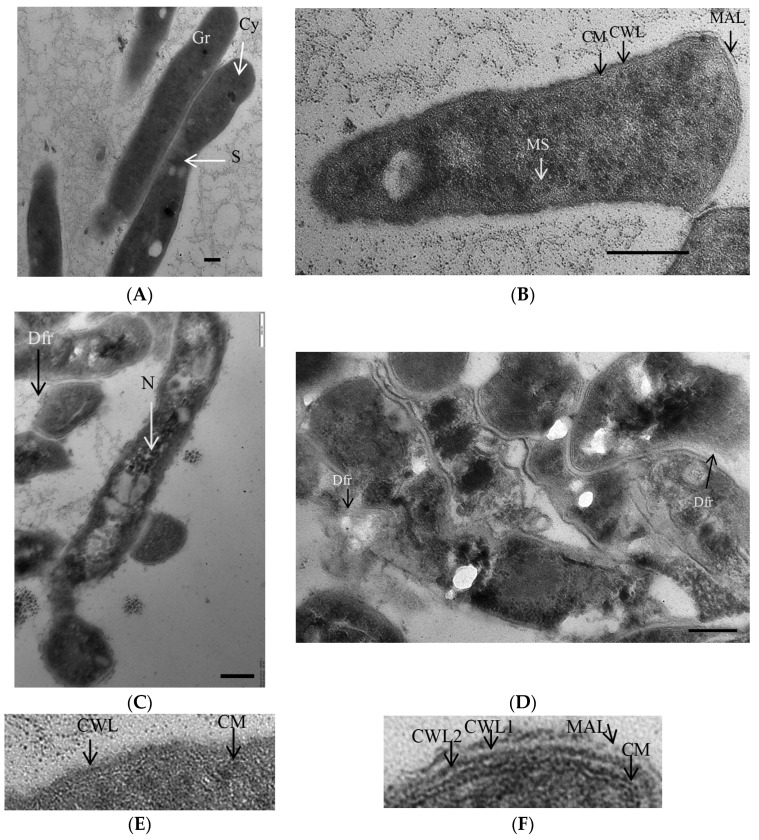
Thin-sectioning TEM images of cells (**A**–**D**) and cell envelope fragments (**E**,**F**) in the control active (**A**,**B**,**E**) and K^+^ deficient cultures (**C**,**D**,**F**). Bars to (**A**–**E**), 200 µm. Designations: CW, cell wall; MAL, mycolic acid layer; CWL cell wall layer; CM, cytoplasmic membrane; Cy, cytoplasm; N, nucleoid; MS, membrane-like structures; S, septum; Gr, granules; Dfr, defragmented cell wall.

**Figure 4 microorganisms-11-02690-f004:**
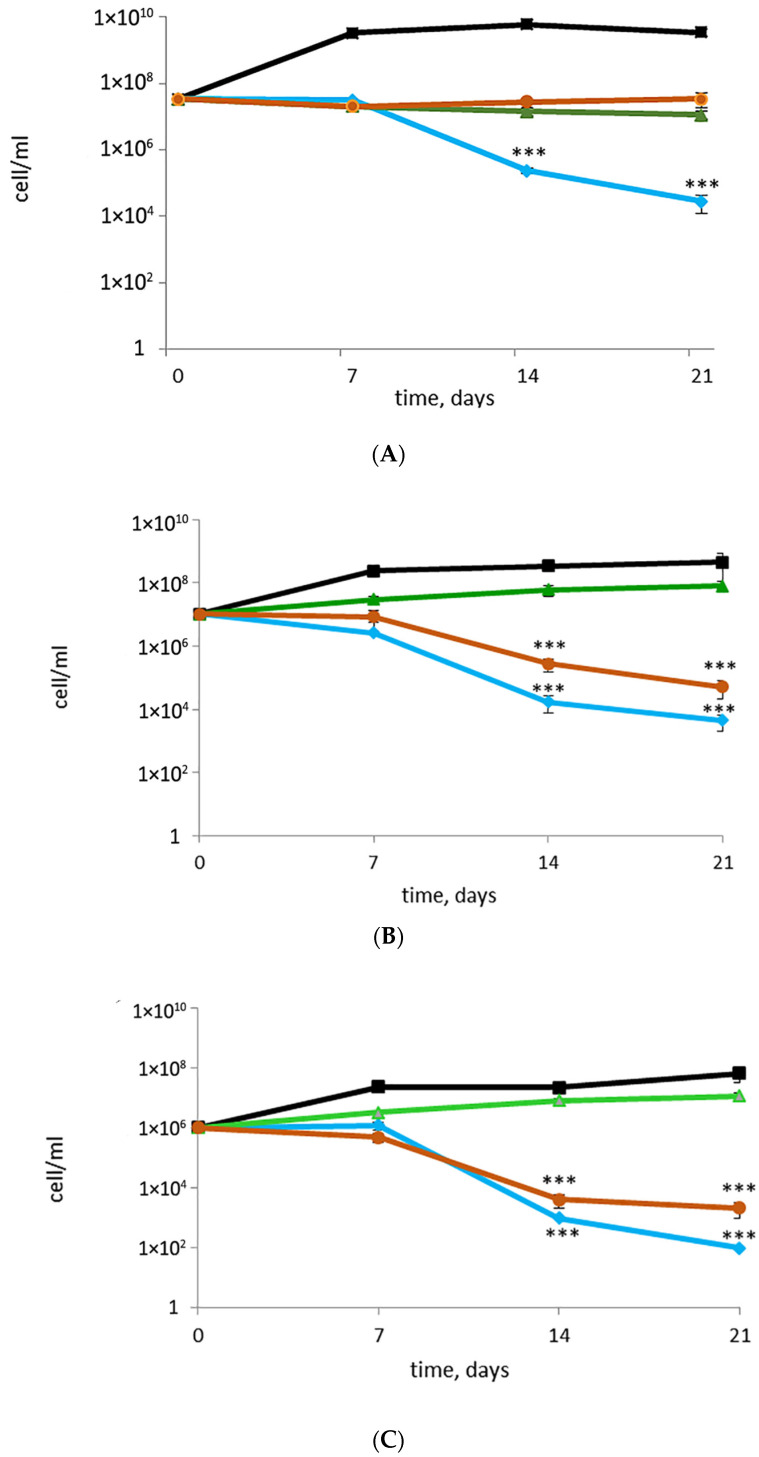
Effects of antibacterials on *M. abscessus*: CFU counts: (**A**) actively growing cells; (**B**) 24-day-old ‘non-culturable’ cells under K^+^ deficiency; (**C**) 44-day-old ‘non-culturable’ cells under K^+^ deficiency. Black line—no-drug control, green line—RIF, 200 µg/mL, maroon line—BDQ, 200 µg/mL, blue line—Moxi, 100 µg/mL. *** *p* < 0.001, unpaired *t*-test.

**Figure 5 microorganisms-11-02690-f005:**
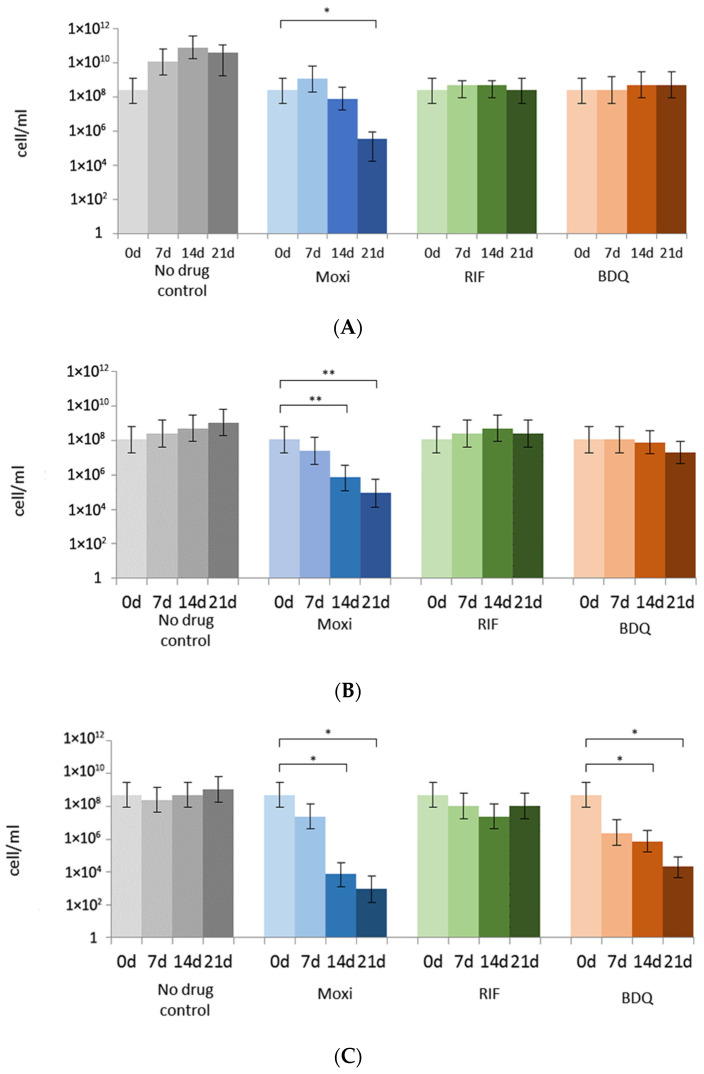
Effects of antibacterials on *M. abscessus*: MPN counts: (**A**) actively growing cells; (**B**) 24-day-old ‘non-culturable’ cells under K^+^ deficiency; (**C**) 44-day-old ‘non-culturable’ cells under K^+^ deficiency. * *p* < 0.05, ** *p* < 0.01, unpaired *t*-test.

## Data Availability

Data are contained within the article.

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
