# Peer review of "Distinct Effects of Moxifloxacin and Bedaquiline on Growing and ‘Non-Culturable’ Mycobacterium abscessus"

_microorganisms, 2023, doi:10.3390/microorganisms11112690_

Round 1

Reviewer 1 Report

Comments and Suggestions for Authors

The manuscript titled “Distinct effects of moxifloxacin and bedaquiline on growing and non-culturable Mycobacterium abscessus” aimed to investigate the capability of M. abscessus to enter metabolically inactive state under K+ deficiency and to compare the effects of antibiotics on active and dormant cells. This is a well-written article and I anticipate that the manuscript should be of great interest to the researchers working on microbial culturing, bactericidal activity and drug tolerance of microbes. I considered the manuscript suitable for publication subject to following improvements.

General comments

Overall, the manuscript is well designed and presented in a good way. However, some modifications are needed to improve the article for publication.

Specific Comments

I would suggest adding novelty of and background of the study in the abstract section.

The abstract section is well written. However, some significant results should be added in the said section.

Reference number 1 is cited five times in the first paragraph. Can you revise it?

The introduction part is well written but need more information (regarding clinical data of M. abscessus) to elaborate.

Revise the objective of the study and elaborate.

Kindly add information regarding statistical software you used in the present analysis.

Appropriate and adequate references to related works are covered sufficiently in the list. Incorporate some latest reference.  

The article should become acceptable after minor revisions of English / grammatical mistakes.

Author Response

Dear Reviewer, thank you for your comments. We much appreciate your time and effort you have put to review the manuscript and to give us helpful comments. We have carefully reviewed all the comments and thoroughly revised the manuscript. All changes are highlighted with yellow in the revised manuscript.

The manuscript titled “Distinct effects of moxifloxacin and bedaquiline on growing and non-culturable Mycobacterium abscessus” aimed to investigate the capability of M. abscessus to enter metabolically inactive state under K+ deficiency and to compare the effects of antibiotics on active and dormant cells. This is a well-written article and I anticipate that the manuscript should be of great interest to the researchers working on microbial culturing, bactericidal activity and drug tolerance of microbes. I considered the manuscript suitable for publication subject to following improvements.

General Comments

Overall, the manuscript is well designed and presented in a good way. However, some modifications are needed to improve the article for publication.

Thank you very much for the positive evaluation.

Specific Comments

I would suggest adding novelty of and background of the study in the abstract section.

The abstract section is well written. However, some significant results should be added in the said section.

Thank you for this comment. We added the novelty and background in Abstract, as well as some important results. Hopefully, now this section looks better.

Reference number 1 is cited five times in the first paragraph. Can you revise it?

We revised it according to your suggestion.

The introduction part is well written but need more information (regarding clinical data of M. abscessus) to elaborate.

We amended the Introduction with some information about treatment of M. abscessus-caused infections, according to Reviewer’s suggestion.

Revise the objective of the study and elaborate.

Thank you for this proposal. We tried to revise the objective and to make it clearer and comprehensive than in the previous version as below.

“The main objective of this study was to assess the efficacy of antibiotics to be efficient against active and dormant Mycobacterium abscessus cells. To this aim, we investigated the capability of this bacterium to enter metabolically inactive state under the deficiency of potassium ions in the growth media and focused on developing a model of non-replicating cells as a useful tool for screening of antibacterial agents. This study included testing a combined approach to monitor the residual viability of both active and dormant M. abscessus cells after exposure to antibacterial agents screened for a reliable evaluation of their bacteriostatic and bactericidal effects. Finally, we attempted to compare the efficacy of several antibiotics, particularly anti-tuberculosis agents, with respect to actively growing and non-replicating M. abscessus.”

Kindly add information regarding statistical software you used in the present analysis.

Thank you for this comment. We added information about statistical software.

Appropriate and adequate references to related works are covered sufficiently in the list. Incorporate some latest reference.  

Some latest references have been added (doi: 10.3390/microorganisms11010090, doi: 10.1016/j.cmi.2023.08.036, doi.org/10.1136/thoraxjnl-2015-207360, doi: 10.1183/13993003.00535-2020, doi: 10.3390/antibiotics9010018)

The article should become acceptable after minor revisions of English / grammatical mistakes.

We checked the text several times and tried to correct mistakes as much as possible.

Reviewer 2 Report

Comments and Suggestions for Authors

The aim of the manuscript is to show the M. abscessus’s capability to enter metabolically inactive state under K+ deficiency and to compare the effects of antibiotics on active and dormant cells.

It was clearly shown that those cells showed a reduced ability to form colonies on nutrient agar, while the MPN (most probable number of viable cells) values were stable, indicating that cells remained viable and can grow in the standard liquid medium.

Concerning the thermal resistance profile for old culture with CFU and MPN dropping after heating at 70°C and further increase (Fig. 2C), the explanation of heterogeneity of population to heat stress must be reconsidered. Actual numbers of CFU and MPN for 44-day-old non-culturable type of cells are very similar to those found for 24-day-old non-culturable cells at 80ºC. So, idea that most cells were eliminated at 70ºC whereas a minor fraction escaped from the detection with viability tests and needed elevated temperatures to activate the growth is not supported.

Differential effect of moxifloxacin and bedaquiline on actively growing and non-culturable M. abscessus, is clearly shown.

And I agree with authors that for estimation of efficacy of drugs in Mycobacterium it is necessary to monitor the ability of cell growth using MPN assay since a subpopulation that cannot be cultured on agar media but regain the growth in the liquid media may exist.

Many references lack the page numbers.

Comments on the Quality of English Language

 Minor editing of English language would be required

Author Response

Dear Reviewer, thank you for your comments. We much appreciate your time and effort you have put to review the manuscript and to give us helpful comments.  We have carefully reviewed all the comments and thoroughly revised the manuscript. All changes are highlighted with yellow in the revised manuscript.

The aim of the manuscript is to show the M. abscessus’s capability to enter metabolically inactive state under K+ deficiency and to compare the effects of antibiotics on active and dormant cells.

It was clearly shown that those cells showed a reduced ability to form colonies on nutrient agar, while the MPN (most probable number of viable cells) values were stable, indicating that cells remained viable and can grow in the standard liquid medium.

Thank you very much for the positive evaluation.

Concerning the thermal resistance profile for old culture with CFU and MPN dropping after heating at 70°C and further increase (Fig. 2C), the explanation of heterogeneity of population to heat stress must be reconsidered. Actual numbers of CFU and MPN for 44-day-old non-culturable type of cells are very similar to those found for 24-day-old non-culturable cells at 80ºC. So, idea that most cells were eliminated at 70ºC whereas a minor fraction escaped from the detection with viability tests and needed elevated temperatures to activate the growth is not supported.

Thank you for this comment. This was our suggestion, and this idea needs further corroboration, despite we have previously observed a similar pattern with dormant myco- and other bacteria. We decided to remove this questionable information as being not too relevant to the major results.

Differential effect of moxifloxacin and bedaquiline on actively growing and non-culturable M. abscessus, is clearly shown.

And I agree with authors that for estimation of efficacy of drugs in Mycobacterium it is necessary to monitor the ability of cell growth using MPN assay since a subpopulation that cannot be cultured on agar media but regain the growth in the liquid media may exist.

Many references lack the page numbers.

We thank the reviewer for noticing this oversight. All references in the list have been double checked and corrected.

 Minor editing of English language would be required

We checked the text several times and tried to correct mistakes as much as possible.